# INTERCEPT Pathogen Reduction in Platelet Concentrates, in Contrast to Gamma Irradiation, Induces the Formation of *trans*-Arachidonic Acids and Affects Eicosanoid Release during Storage

**DOI:** 10.3390/biom12091258

**Published:** 2022-09-07

**Authors:** Gerda C. Leitner, Gerhard Hagn, Laura Niederstaetter, Andrea Bileck, Kerstin Plessl-Walder, Michaela Horvath, Vera Kolovratova, Andreas Tanzmann, Alexander Tolios, Werner Rabitsch, Philipp Wohlfarth, Christopher Gerner

**Affiliations:** 1Department of Blood Group Serology and Transfusion Medicine, Medical University of Vienna, 1090 Vienna, Austria; 2Department of Analytical Chemistry, Faculty of Chemistry, University of Vienna, 1090 Vienna, Austria; 3Joint Metabolome Facility, Faculty of Chemistry, University of Vienna, 1090 Vienna, Austria; 4Division of Biomedical Science, University of Applied Sciences, FH Campus Wien, 1100 Vienna, Austria; 5Internal Medicine 1, Stem Cell Transplantation Unit, Medical University of Vienna, 1090 Vienna, Austria

**Keywords:** eicosanoids, high-resolution mass spectrometry, liquid chromatography, pathogen reduction, platelet concentrates, platelets, *trans*-arachidonic acids

## Abstract

Pathogen inactivation techniques for blood products have been implemented to optimize clinically safe blood components supply. The INTERCEPT system uses amotosalen together with ultraviolet light wavelength A (UVA) irradiation. Irradiation-induced inactivation of nucleic acids may actually be accompanied by modifications of chemically reactive polyunsaturated fatty acids known to be important mediators of platelet functions. Thus, here, we investigated eicosanoids and the related fatty acids released upon treatment and during storage of platelet concentrates for 7 days, complemented by the analysis of functional and metabolic consequences of these treatments. Metabolic and functional issues like glucose consumption, lactate formation, platelet aggregation, and clot firmness hardly differed between the two treatment groups. In contrast to gamma irradiation, here, we demonstrated that INTERCEPT treatment immediately caused new formation of *trans*-arachidonic acid isoforms, while 11-hydroxyeicosatetraenoic acid (11-HETE) and 15-HETE were increased and two hydroperoxyoctadecadienoic acid (HpODE) isoforms decreased. During further storage, these alterations remained stable, while the release of 12-lipoxygenase (12-LOX) products such as 12-HETE and 12-hydroxyeicosapentaenoic acid (12-HEPE) was further attenuated. In vitro synthesis of *trans*-arachidonic acid isoforms suggested that thiol radicals formed by UVA treatment may be responsible for the INTERCEPT-specific effects observed in platelet concentrates. It is reasonable to assume that UVA-induced molecules may have specific biological effects which need to be further investigated.

## 1. Introduction

One of the key elements of blood transfusion is the availability of safe blood and blood products. In addition to the careful and sophisticated donor selection, application of state-of-the-art blood processing procedures and testing is mandatory.

Transfusion-associated graft-versus-host disease (TA-GvHD) is a very rare but usually fatal complication following transfusion of lymphocyte-containing blood components in immunosuppressed patients [1,2,3]. Furthermore, non-immunosuppressed patients could experience this complication, particularly if the transfused blood components are not donated from an HLA-haploidentical unrelated donor or family member [4,5,6,7,8]. Since the early 1980s, irradiation of blood components of at least 25 Gy is strongly recommended for patients at risk of developing TA-GvHD [9]. Gamma irradiation inactivates residual T cells in blood components, the source cells for TA-GvHD. However, gamma irradiation (γ-IRR) may induce metabolic injuries and reduces the shelf life of blood components [10].

In recent decades, large efforts have been made to further increase blood safety and reduce the risk of transfusion-transmitted infections (TTIs). The most recent innovation in this field was the development of pathogen reduction (PR) methods and inactivation (PI) techniques for platelet concentrates (PCs) and plasma. These techniques are effective against proliferating nucleated cells of pathogens, and, thus, also effective against TA-GvHD-inducing T cells. Thus, irradiation of blood components in patients at risk can be omitted. Several studies have been carried out to prove the hemostatic efficacy of PR or PI platelets [11,12,13]. However, while some studies have reported that post-transfusion increment of platelet counts was significantly lower after transfusion of PR and PI platelets, no increase in clinically relevant bleeding was observed [14,15]. Anyhow, concerns about the function of these pretreated platelets arose. In a previous proteomic investigation, it was stated that gamma irradiation resulted in the acceleration of platelet storage lesions, and the Mirasol treatment only moderately exacerbated these phenomena [16].

These observations encouraged us to examine the molecular level in relation to γ-IRR and INTERCEPT treatment by measuring the metabolic changes and the formation of eicosanoids.

Polyunsaturated fatty acids are highly likely candidate molecules present in platelets as they may show chemical reactivity upon irradiation [17]. Arachidonic acid (AA) is a polyunsaturated fatty acid (PUFA) abundantly esterified in membrane lipids such as phosphatidylcholines. Oxidation of AA may occur in an enzymatic or nonenzymatic fashion, eventually forming so called eicosanoids, which are known to be of high functional relevance for platelets [18]. The so-called cyclooxygenase pathway, the cytochrome p450 pathway, and the lipoxygenase pathway, each comprising several enzymes with distinct substrate and product specificity, are responsible for the formation of a great variety of molecules [19]. However, other PUFAs, such as linoleic acid, may be processed by these enzymatic pathways, further increasing the complexity of the family of fatty acid-derived oxylipins. While many eicosanoids are known as the main mediators of inflammation-associated clinical symptoms, thromboxane A2 is formed upon platelet activation and is known as the main prostanoid further promoting platelet aggregation, which is further modulated by various prostaglandins and hydroxyeicosatetraenoic acids (HETEs) [20]. Due to their intrinsic complexity, the influencing role of most eicosanoids on coagulation has not yet been fully established. However, it is evident that the eicosanoid background may significantly affect the coagulation properties of transfused platelet concentrates [21]. Thus, here, we investigated the immediate effects and storage effects of γ-IRR or INTERCEPT treatment with regard to eicosanoids and the related molecules using an untargeted high-resolution mass spectrometry-based method as employed by us previously [22].

## 2. Materials and Methods

### 2.1. Study Design

Fifteen double-unit PCs were collected from 15 healthy volunteers of our multicomponent donor pool. All the donors provided written informed consent to participate in this study. All the PCs were divided into two single units so that the requirements for PI of a single unit were fulfilled. The second smaller unit was transferred into an oxygen-permeable storage bag from Terumo BCT (Denver, CO, USA). One unit was pathogen-inactivated and one unit was irradiated with 30 Gy. The PCs were stored at 22 °C (±2 °C) under continuous agitation on an agitator (Helmer Platelet Incubator, Noblesville, IN, USA) for 7 days. Samples were drawn under aseptic conditions after production (day 1, before dividing into two parts), after either γ-IRR or INTERCEPT treatment (day 2), further on days 5 and 7 (end of shelf life). All the units were tested for platelet counts, residual white blood cells (rWBC), pH, lactate, lactate dehydrogenase (LDH), glucose, and thromboelastometry (TEM) measurements. TEM was performed to work out if both manipulations have an impact on the hemostatic function of fresh and stored collected platelets or not. Sterility testing was performed only on day 7. Additionally, all the units were tested for the aggregation response to collagen and thrombin receptor agonist peptide 6 (TRAP-6). PI treatment was performed with an Intercept™ Blood System (Cerus Europe B.V., Amersfoort, The Netherlands). All the PCs were visually inspected for swirling and aggregates.

### 2.2. Donors

Collection of apheresis-derived single-donor platelet units is a core competence of the Department of Blood Group Serology and Transfusion Medicine at the Medical University of Vienna. All the scheduled donors (11 men, three women) met the national and international requirements for healthy volunteers (Austrian law, EU guidelines); the median age was 39 years (range, 22–65). Double adult dose PCs were collected using a TrimaAccel collection system (version 6.0; Terumo BCT, Denver, CO, USA).

### 2.3. Clinical Laboratory Analyses

Platelet counts were measured with a hematology analyzer (Cell Dyn Ruby, Abbott, South Pasadena, CA, USA); pH was measured with a biochemistry analyzer at 22 °C (ABL 80 FLEX Analysator, Drott, Medizintechnik GmbH, Wiener Neudorf, Austria); LDH was analyzed with Beckmann Coulter AU 5800 (Beckman Coulter GmbH, Bremerhaven, Germany), glucose and lactate—with OLYMPUS AU 5430 (Beckmann Coulter, Danvers, MA, USA).

#### 2.3.1. Platelet Function and Activation by Light Transmission Aggregometry (LTA)

Platelet function and activation by LTA was performed using PAP 8E (MoeLab, Langenfeld, Germany) as previously described [23,24]. In short, the PCs were diluted to a final concentration of 250 × 10^9^/L with solvent-detergent plasma (Octaplas; blood group AB; Octapharma, Vienna, Austria). The aggregation response to collagen (final concentration, 190 µg/mL; BioData Corporation, Horsham, PA, USA) or TRAP-6 (final concentration of 25 µmol; Bachem, Bubendorf, Switzerland) was continuously recorded for 5 min. The maximal aggregation (MA) was used for further statistics.

#### 2.3.2. Thromboelastometry

ROTEM^®^ (Pentapharm GmbH, Munich, Germany) is a point-of-care analyzer which uses normally citrated whole blood samples for the analysis of viscoelastic properties of a sample as it clots. Here, the PC samples were treated similarly to LTA measurements to mimic whole blood. In our department, only the NATEM test was performed. In this test, clot formation is initiated through the activation of coagulation by contact. The clot formation time (CFT) and the maximal clot firmness (MCF) were regarded as the primary variables of interest, as both are mainly influenced by the number and the function of platelets [25,26]. The CFT is defined as the time from the start of clotting until the clot firmness reaches the 20 mm mark. The MCF is the greatest vertical amplitude of a typical trace and reflects the absolute strength of a fibrin and platelet clot [27].

#### 2.3.3. Residual Leukocytes and Sterility Tests

Residual leukocytes were assessed only on the first day of storage by flow cytometry using a Leucocount kit (Becton Dickenson, San Jose, CA, USA). Sterility tests were performed by an accredited institute according to the requirements of Ph.Eur. 2.

#### 2.3.4. Pathogen Inactivation of Platelets

Our institution produces solely single-donor platelets by apheresis. We have been using the Intercept™ system for PI of PCs since March 2012. This system has been shown to effectively inactivate contamination of PCs with a broad spectrum of viruses, bacteria, and parasites as well as donor leukocytes. Austrian authorities allow 7-day storage for 100% PI PCs, which has therefore been implemented as the maximum storage time at our institution. The Intercept™ system uses amotosalen HCl (a photoactive compound) and long-wavelength ultraviolet (UVA) illumination for *ex vivo* PI of PCs. Residual amotosalen and free photoproducts are reduced to low levels by exposure to a compound adsorption device (CAD) before transfer of the treated platelets to a storage container for release.

#### 2.3.5. Irradiation of Platelets

The 30 Gy irradiation of the second unit of the PCs was performed with IBL 437 C (CIS Bio international, Saclay, France). This device uses a cesium-137 source, representing a common technique for irradiation of blood products.

### 2.4. Sample Preparation for Eicosanoid and Fatty Acids Analyses

An aliquot of platelet concentrate (1 mL) was transferred into a 15 mL Falcon tube for two different treatment types. Ice cold ethanol (EtOH) (4 mL) was added, containing 10–100 nM of each internal standard (12S-hydroxyeicosatetraenoic acid (HETE)-d8, 15S-HETE-d8, 5-oxo-eicosatetraenoic acid (ETE)-d7, 11,12-dihydroxy-5Z,8Z,14Z-eicosatrienoic acid (DiHETrE)-d11, prostaglandin E2 (PGE2)-d4, 20-HETE-d6 (Cayman Chemical, Tallinn, Estonia), for protein precipitation over night at −20 °C. The samples were then centrifuged for 30 min at 4536× *g* and 4 °C, protein pellets were discarded, and EtOH was evaporated *via* vacuum centrifugation at 37 °C until the original sample volume was restored. Solid-phase extraction (SPE) was performed with 30 mg/mL StrataX SPE columns (Phenomenex, Torrance, CA, USA). The columns were washed with 2 mL absolute methanol (abs. MeOH) equilibrated with 2 mL MS grade H_2_O before sample loading with Pasteur pipettes. The samples were washed with 5 mL MS grade H_2_O before eluting the eicosanoids into 1.5 mL glass vials with 500 µL abs. MeOH containing 2% formic acid (FA). The eluates were stored at −80 °C until further analysis. Before measurement, the organic phase was evaporated at room temperature using nitrogen steam and reconstituted in 150 µL reconstitution buffer (H_2_O/acetonitrile (ACN)/MeOH + 0.2% FA—65:31.5:3.5).

### 2.5. Synthesis of trans-Arachidonic Acids (trans-AAs)

*Trans*-AAs were synthesized via isomerization of all the *cis*-arachidonic acids according to Ferreri et al. [28], applying a slightly adapted protocol. In brief, a solution of 1.5 mM arachidonic acid in isopropanol containing 10 mM β-mercaptoethanol was placed in an iOS quartz cuvette and bubbled with a gentle stream of nitrogen for 75 min, while exposing to 364 nm wavelength (UVP UVGL-58, Analytik Jena, Jena, Germany). Afterwards, isopropanol and β-mercaptoethanol were evaporated using nitrogen and the sample reconstituted in 150 µL reconstitution buffer containing the internal standards as described above.

### 2.6. LC–MS/MS Analysis

Separation of the analytes was achieved with a Vanquish™ UHPLC system (Thermo Fisher Scientific, Vienna, Austria) equipped with a reversed-phase C18 column (Kinetex^®^, 2.6 µm XB-C18, 100 Å, LC column, 150 × 2.1 mm, Torrance, CA, USA). The flow rate was set to 200 µL min^−1^, the HPLC oven temperature was set to 40 °C, the autosampler was set to 4 °C, and the injection volume was 20 µL. A gradient method with a total run time of 20 min was applied using eluent A (H_2_O + 0.2% FA) and eluent B (90% ACN + 10% MeOH + 0.2% FA). The starting condition of 35% B was kept constant for 1 min, increasing linearly up to 90% B within 9 min and then further increasing up to 99% B within 0.5 min, where it was kept constant for 5 min, restoring the initial conditions within 0.5 min.

MS analysis was performed using a high-resolution quadrupole orbitrap mass spectrometer (Thermo Scientific™ QExactive™ HF hybrid quadrupole orbitrap mass spectrometer) equipped with an HESI source run in the negative ionization mode. Data were recorded in the full-scan mode operating in the mass range of *m*/*z* 250–700 at the resolution of 60,000 (*m*/*z* 200). A Top 2 method was chosen for MS/MS fragmentation in the data-dependent acquisition (DDA) mode, and fragmentation was achieved using an HCD with a normalized collision energy of 24 at a resolution of 15,000 (*m*/*z* 200). Spray voltage was set to 3.5 kV, capillary temperature was set to 253 °C, sheath gas and auxiliary gas were set to 46 and 10 arbitrary units, respectively.

### 2.7. LC–MS/MS Data Processing

Identification of eicosanoids was performed manually using Xcalibur Qual Browser (version 4.1.31.9; Thermo Fisher Scientific, Vienna, Austria) by using the exact mass, retention time, and MS/MS fragmentation pattern which were manually compared to the reference spectra in the LIPIDMAPS depository library from July 2020 [29]. These putatively identified molecules are marked with their retention time after either the exact mass or MS2-based proposed identification. External standards were then used for verification of the putatively identified fatty acids. In case of the presently described hydroperoxyoctadecadienoic acids (HpODEs), monoisotopic exact mass, isotopic pattern, and fragmentation spectra clearly identified the molecular lipid, but did not allow us to define the molecular structure. However, a comparison with commercial standards ruled out 9-HpODE and 13-HpODE, respectively. A lack of further commercially available standards did not yet allow us further clarification. For relative quantification, TraceFinder (version 4.1; Thermo Fisher Scientific, Vienna, Austria) was applied, allowing a mass deviation of 5 ppm. The resulting peak areas of each analyte were normalized to the mean peak area of all the internal standards within the corresponding sample, giving normalized area-under-the-curve (nAUC) values. This also allows for the correction of variances arising from sample extraction and LC–MS/MS analysis. For visualization and statistical evaluation, GraphPad Prism (version 6.07, GraphPad Software, San Diego, CA, USA) and Origin Pro (version 2019b, OriginLab Corporation, Northampton, MA, USA) were used. In addition, raw data were read using the R software package (version 4.2.0) [30]. The data were log2-transformed. For normalization, the log2-transformed mean area of the internal standards was subtracted from the log2-transformed analyte areas. To obtain values similar to proteomics LFQ values (thus enabling imputation), 20 was added to the log2-transformed areas. Missing values were imputed using the minProb function of the imputeLCMD package (version 2.1) [31]. Volcano plots were produced using the EnhancedVolcano package (version 1.14.0) [32].

### 2.8. Statistical Analysis

If not stated otherwise, the results are shown as the means and standard deviations (SD) or the medians and ranges for descriptive purposes. All the statistical comparisons were conducted using nonparametric tests due to the nonnormal distribution of data. Differences between the two groups (irradiated and pathogen-inactivated PCs) and changes during storage were analyzed using the Mann–Whitney U test. For the evaluation of metabolic changes within the groups over time, the Wilcoxon paired signed-rank test for dependent samples was applied. A two-sided *p*-value < 0.05 was considered statistically significant.

## 3. Results

Platelet concentrates were prepared and subjected to γ-IRR or INTERCEPT treatment according to established protocols as outlined in the Materials and Methods. Untreated controls were harvested and analyzed at day 1. At days 2, 5 and 7, aliquots of each treatment option were analyzed either with regard to clinical parameters or using LC–MS/MS to determine eicosanoids.

### 3.1. Platelet Metabolism and Function Are Hardly Affected Either by Gamma Irradiation or INTERCEPT Treatment

Glucose consumption may serve as an indicator for platelet metabolism and was evidenced by a steady decrease in glucose concentration accompanied by the formation of lactate during storage of platelet concentrates (Figure 1A,B). A comparison between the two treatment options pointed to a slightly decreased formation of lactate on day 7 upon INTERCEPT treatment. Remarkably, clot formation time and clot firmness as well as the TRAP-induced maximal aggregation did not differ significantly between these two groups. However, collagen- and TRAP-6-induced aggregation decreased significantly over time (*p* < 0.05), whereas clot formation time and clot firmness remained stable during the storage time (Table 1, Figure 1C,D).

### 3.2. INTERCEPT Treatment for Pathogen Reduction Induces the Specific Formation of trans-Arachidonic Acids (trans-AAs) and Also Affects Other Eicosanoids

Pathogen reduction using INTERCEPT treatment had immediate effects on eicosanoids and their precursor, arachidonic acid. Figure 2A shows the treatment-induced specific formation of isoforms of arachidonic acid, which were identified by means of fragmentation analysis (MS2 spectra) as various forms of *trans*-AA; *trans*-AAs were otherwise reported to be formed either by the action of nitric oxide radicals *in vivo* [31] or by thiol radicals generated with radical reaction initiators such as azide radicals [28]. In addition, several eicosanoids such as 11-HETE, 15-HETE, and 15-deoxy-PGJ2 were found significantly upregulated upon UVA treatment (Figure 2B, Appendix A). On the other hand, two different isoforms of HPODE were found uniformly downregulated, as exemplified in Figure 2C. While INTERCEPT treatment was associated with such robust and uniform regulatory events, γ-IRR did not induce any similarly striking events at the levels of eicosanoids or other fatty acids.

### 3.3. UVA Treatment Is Capable of Forming trans-Arachidonic Acids in the Presence of a Sulhydryl Donor

The formation of *trans*-AAs in platelet concentrates upon INTERCEPT treatment was unexpected. The known mechanisms for the generation of *trans*-AAs pointed to the involvement of radicals. Whether and how INTERCEPT treatment might form radicals is not known. Thus, we wondered whether UV light might be capable of initiating the formation of sulfhydryl radicals in the absence of amotosalen or other potential radical reaction initiators. Under a constant flow of nitrogen gas in order to remove any solubilized oxygen, pure AA was irradiated with the same dose of UVA as used for INTERCEPT treatment in the presence of sulfhydryl donor β-mercaptoethanol. Remarkably, *trans*-AA isoforms I and II were indeed formed in this cell-free experiment as observed by INTERCEPT treatment of platelet concentrates (Figure 3). Other eicosanoids were not detected in this experimental setup, pointing to the essential role of enzymes for their formation. Thus, here, we present strong evidence for the notion that UVA treatment is capable of forming sulfhydryl radicals in the presence of suitable sulfhydryl donors, which are apparently capable of forming *trans*-arachidonic acid.

### 3.4. Stored Platelet Concentrates Still Show Significant Differences in Their Eicosanoid Profile

The INTERCEPT treatment-induced alterations of eicosanoids apparently remained stable and were still detectable on day 7 (Figure 4A). Remarkably, lipoxygenase products 12-HETE/8-HETE and 12-HEPE were also detected as significantly downregulated upon INTERCEPT treatment when compared to γ-IRR. Time course analyses point to a combination of background effects with treatment-induced effects. Actually, platelet concentrates have been described to constantly release AA during storage together with several lipoxygenase products such as 12-HETE [33]. The present data evidently confirm this observation in case of the γ-IRR samples, but much less in case of the INTERCEPT-treated samples (Figure 4B).

## 4. Discussion

As long as no stable *in vitro* model system for megakaryocyte culture and platelet formation is presented, patients with urgent demands for platelet concentrates have to be treated with samples derived from individual donors and, thus, subject to substantial interindividual variations. All the parameters recorded in the present study with regard to platelet properties showed high degrees of interindividual variations, which may indeed relate to undesirable variations in the clinical outcome of platelet treatment. Importantly, platelet concentrates have to be pretreated to avoid graft-versus-host disease for patients with severe immunosuppressed conditions (e.g., after peripheral blood stem cell or bone marrow transplantation, or in case of newborns with severe combined immunodeficiency). The method of choice to inactivate T cells is 25–30 Gy γ-irradiation (γ-IRR) of the products [9]. Secondly, pathogen inactivation (PI; INTERCEPT) and pathogen reduction (PR) methods were implemented to minimize the risk of transfusion-transmitted infections (TTIs). These methods affect and inactivate nucleated cells and, thus, are also effective for inactivating the candidate cells for GvHD. As a consequence, in case of PI, γ-IRR can be omitted. However, these treatments may induce further biochemical alterations which have not yet been investigated in molecular detail as presented here. Pathogen inactivation methods are becoming increasingly important due to increased risk of TTI. However, a possible negative influence on platelet function in the latter is an ongoing discussion. Here, we tried to investigate objective parameters independent of the patient’s individual condition.

Eicosanoids are a group of lipid-signaling molecules formed by the oxidation of PUFAs [19]. Platelets are known producers of these highly bioactive molecules [33], motivating us to focus on this class of biomolecules. Importantly, eicosanoids are potent modulators of immune functions [34], with direct effects on various leukocyte functions [22]. The association with innate immune functions and various diseases, including cancer [35,36], makes these molecules potentially relevant for chronic diseases.

Here, we demonstrated that INTERCEPT treatment has direct effects, such as the formation of *trans*-AAs. As *tran*-AAs formation has been described to be catalyzed by radicals [28,37], we wondered whether UVA treatment potentially causes some radical formation. Therefore, we exposed cell-free, actually pure arachidonic acid in the absence of oxygen but presence of sulfhydryl donor β-mercaptoethanol to UVA light in the same experimental conditions as applied for INTERCEPT treatment. Following this protocol, we were indeed able to synthesize *trans*-AAs and apparently reproduced the *trans*-AA pattern observed upon INTERCEPT treatment of platelet concentrates (Figure 3). Thus, here, we demonstrated that UVA light in the presence of PUFAs and sulfhydryl groups induced *cis*–*trans* isomerization of carbon double bonds, most likely via the formation of sulfhydryl radicals acting as catalysts. Actually, *trans*-fatty acids may have enzyme affinities and substrate specificities substantially different from the naturally occurring all-*cis* isoforms. Increased levels of *trans*-fatty acids have been implicated with increased risk for cardiovascular diseases [38]. While there is evidence that *trans*-AAs may inhibit NADPH-oxidase activity [39], thus interfering with neutrophil effector functions, potential implications of *trans*-AAs on blood coagulation properties have not yet been described.

In addition to the formation of *trans*-AAs, we have also observed the increased formation of cyclooxygenase 1 (COX1) product 15-deoxy-PGJ2, 15-LOX product 15-HETE, and P450 product 11-HETE upon INTERCEPT treatment (Figure 4). This is in stark contrast to the formation of 12-LOX products 12-HETE and 12-HEPE, which were significantly higher in γ-IRR samples (Figure 4A). These observations strongly indicate UVA induced specific inhibition of the 12-LOX enzyme activity as already described in the case of keratinocytes exposed to sunlight [40].

It has been well-described that specific inhibition of, e.g., COX-1 also impacts the formation of other enzymatic products, such as 11-HETE and 15-HETE, pointing to complex regulatory mechanisms [41]. Actually, we did not observe any direct strong effect of γ-IRR, and the apparent differences in eicosanoid concentrations during the analyzed time course may likely have resulted from some background platelet enzymatic activity during storage. Actually, phospholipase A2 activity is known to cause constant release of PUFAs during platelet storage [42]. These PUFAs are then subject to a broad variety of enzymatic or nonenzymatic oxidation processes, resulting in the formation of oxylipins with remarkable bioactivities [19]. Consumption of the precursor molecule AA to form *trans*-AAs upon INTERCEPT treatment may, thus, also have caused the relatively lower levels of the enzymatic products thereof compared to the γ-IRR samples as displayed in Figure 4.

## 5. Conclusions

The present data demonstrate that INTERCEPT treatment causes formation of *trans*-AAs, hardly interferes with the formation of P450 or COX products (or not at all), but apparently inhibits formation of 12-LOX products. We did not find evidence of a similarly direct effect of gamma irradiation on PUFA processing and eicosanoid formation. Thus, the two treatment options for platelet concentrates are associated with marked differences in eicosanoid formation with potential clinical implications for the recipient, which need to be addressed in future studies.

## Figures and Tables

**Figure 1 biomolecules-12-01258-f001:**
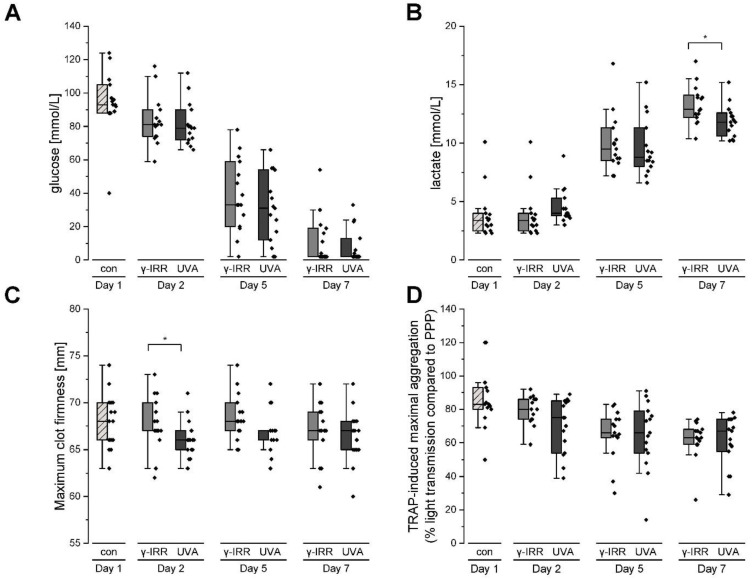
Metabolic changes of platelet concentrates over the storage time for (**A**) glucose and (**B**) lactate. Functional parameters are represented *via* (**C**) the maximal clot firmness (MCF) and (**D**) the maximal TRAP (thrombin activation peptide)-induced aggregation capacity. Significant differences between γ-IRR (grey) and INTERCEPT (dark grey)-treated PCs are labeled by asterisks (Mann–Whitney U test).

**Figure 2 biomolecules-12-01258-f002:**
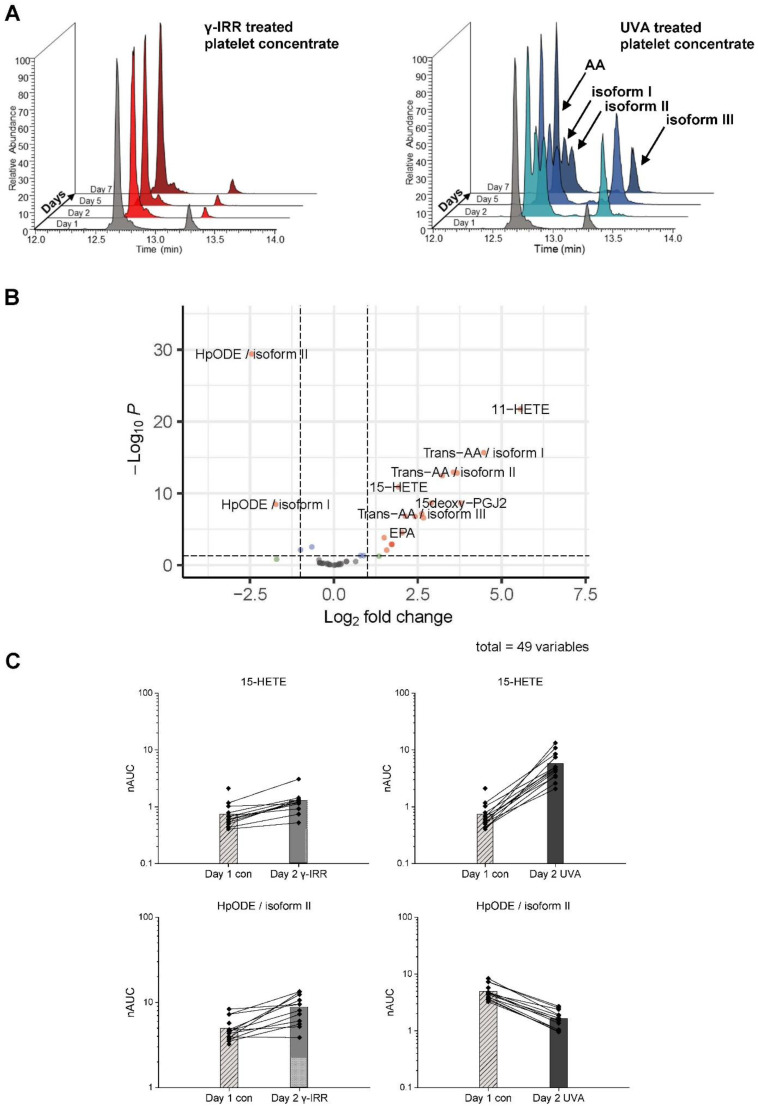
(**A**) INTERCEPT treatment, but not γ-IRR, is accompanied by the formation of *trans*-arachidonic acid isoforms. (**B**) Immediate effects of INTERCEPT treatment versus γ-IRR at day 2 are displayed as a volcano plot: significant regulations with *p*-value < 0.05 and log2 fold change (Log2FC) > |1| (red), *p*-value < 0.05 (blue), Log2FC > |1| (green) and nonsignificant results (grey). (**C**) Uniformity of strong regulatory events upon INTERCEPT treatment in contrast to γ-IRR are shown for 15-HETE and HpODE isoform II.

**Figure 3 biomolecules-12-01258-f003:**
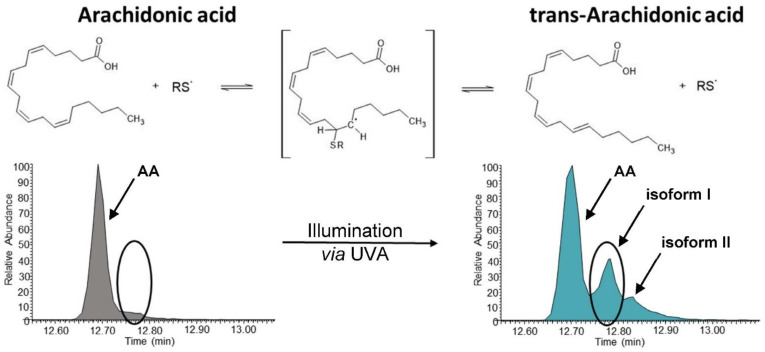
Formation of *trans*-arachidonic acid isoforms I and II obtained by UVA illumination of arachidonic acid in the presence of a sulfhydryl donor (β-mercaptoethanol) in a cell-free experiment.

**Figure 4 biomolecules-12-01258-f004:**
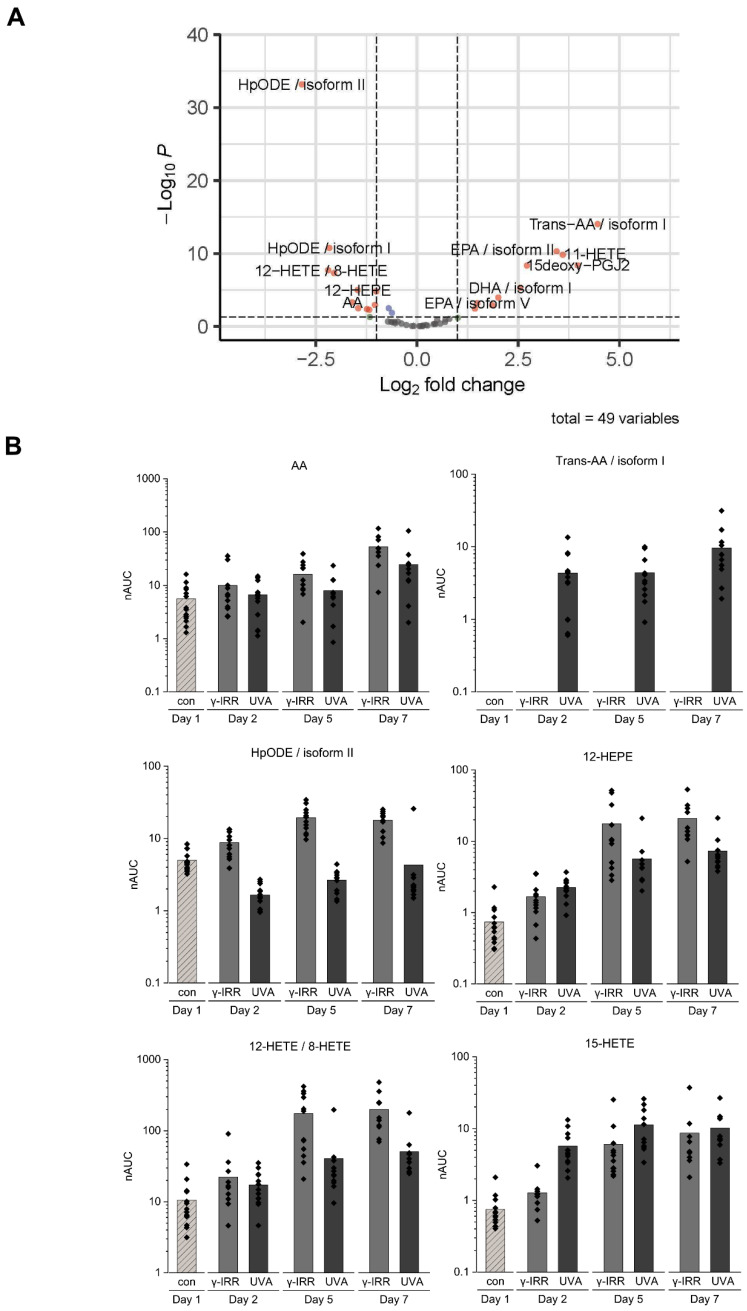
(**A**) Volcano plot of the comparison of γ-IRR and INTERCEPT-treated platelet concentrates on day 7. While the immediate effects observed on day 2 were still detectable, an additional downregulation of 12-LOX products 12-HETE and 12-HEPE was observed. Significant regulations with *p*-value < 0.05 and log2 fold change (Log2FC) > |1| (red), *p*-value < 0.05 (blue), Log2FC > |1| (green) and nonsignificant results (grey). (**B**) Time course analyses of the selected molecules as described in the text.

**Table 1 biomolecules-12-01258-t001:** (**a**): ROTEM data of γ-irradiated and INTERCEPT-treated platelet units. MCF, maximal clot firmness (millimeters); CFT, clot formation time (seconds); SD, standard deviation. (**b**): Aggregometry results after collagen and TRAP-6 enhancement. TRAP-6, thrombin receptor agonist peptide 6; MA, maximal aggregation; SD, standard deviation.

**(a)**				
	**γ-irradiated**		**INTERCEPT-treated**	
	**MCF, mm** **(mean/SD)**	**CFT, s** **(mean/SD)**	**MCF, mm** **(mean/SD)**	**CFT, s** **(mean/SD)**
Day 1	68 (3)	152 (22)	68 (3)	152 (22)
Day 2	68 (3)	162 (33)	66 (1)	156 (29)
Day 5	68 (3)	152 (40)	67 (3)	142 (21)
Day 7	67 (3)	157 *(23)*	65 (2)	173 (27)
**(b)**				
	**γ-irradiated**		**INTERCEPT-treated**	
	**Collagen, MA** **(mean/SD)**	**TRAP-6, MA** **(mean/SD)**	**Collagen, MA** **(mean/SD)**	**TRAP-6, MA** **(mean/SD)**
Day 1	88 (15)	91 (18)	88 (15)	91 (18)
Day 2	81 (7)	81 (6)	81 (10)	69 (17)
Day 5	74 (10)	65 (15)	78 (11)	65 (24)
Day 7	68 (12)	62 (12)	69 (11)	62 (14)

## Data Availability

The data presented in this study are available in Appendix A.

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
