# Peer review of "INTERCEPT Pathogen Reduction in Platelet Concentrates, in Contrast to Gamma Irradiation, Induces the Formation of trans-Arachidonic Acids and Affects Eicosanoid Release during Storage"

_biomolecules, 2022, doi:10.3390/biom12091258_

Round 1

Reviewer 1 Report

A carefully articulated manuscript. Authors are advised to perform a spellcheck.

Author Response

We thank the reviewer for the very positive review!

Reviewer 2 Report

Please see attached WORD document

Author Response

We would like to thank reviewer 2 for excellent and very helpful comments. As outlined below, we actually followed all suggestions, i.e. changed the Title, re-structured the Results section and Figures and tried to improve the presentation of the study aims.

Ad Major remarks:

we have changed the Title to better fit the purpose of the study

Ad General:

We apologize for being unclear. We have included a more comprehensive explanation of the background and meaning of the present study.

Ad Introduction:

As mentioned above, we have now included more explanations and references and presented the project aims with more details.

Ad Materials: We apologize for being unclear. Actually, we did not perform any analysis at day 0 and thus avoided to refer to become clearer. We have corrected all related references.

Ad statistics: we chose the statistical analysis methods in accordance with the requirements and data properties, the local Statistics Department approved the applied methods.

Ad Results: We apologize for mislabeling of the Table and corrected all errors accordingly. The choice of panels was done in order to provide the best possible data clarity, all data are actually shown either visually or in the table.

Values in 1C were corrected as mentioned above

also this error was corrected, only one bar is shown as requested (more than justified!)

Actually also this request is highly justified, we have nor used the same software to produce all Figures, we hope it looks better now.

Ad Figures 2 and 4

We were very happy to hear your really excellent and helpful suggestion, which we realized resulting in a new structure of the Figures. We thank the reviewer again for this great help!

Discussion

Following the suggestions of this reviewer, we have also reformulated the Discussion section accordingly.

The notion regarding a lack of gamma-IRR effects was now included in the Results section

We explained the in vitro synthesis experiment in more detail and omitted any speculation. Actually  we have now included Volcano plots and thus focus more on relative alterations in contrast to the newly formed molecules. As 16-HETE and 18-HETE were actually close to the limit of detection, we decided to omit stressing these results (all data including 16-HETE and 18-HETE are completely contained in Supplementary Table 1).

Ad Abstract:

Also the Abstract was rewritten accordingly.

Reviewer 3 Report

Authors reported that “INTERCEPT” pathogen reduction of platelet concentrates induces trans-arachidonic acids and affects eicosanoid formation.

Although this manuscript is potentially interesting, several issues arise.

The full name of “INTERCEPT” may be helpful for the readers.

ABSTRACT. Results section may be insufficient and unclear.

            Conclusion is recommended to be reconsidered.

            The full name of UVA may be helpful.

What do the “trans arachidonic acids” or other-UVA-induced molecules effect on the recipients?  

Figure 1. The two groups should be explained in legend. The differences of two bars are not clear. Open bar and closed bar may be helpful for the readers.

Table 1b. What are unit of data?

         Statistical analysis between 2 groups may be helpful.

Figure 2 What is “nAUC” and its unit?

Figure 2B recommended to be remade.

Author Response

We thank the reviewer for helpful comments, which improved our manuscript. Please find enclosed a point to point reply to all issues raised.

The full name of “INTERCEPT” may be helpful for the readers.

INTERCEPT™ is a company name, we are afraid we are not aware this might be an abbreviation.

ABSTRACT. Results section may be insufficient and unclear.

We followed the suggestions of reviewer 3 and restructured the results section in order to improve clarity

Conclusion is recommended to be reconsidered.

            The Conclusion was rewritten as requested.

The full name of UVA may be helpful.

            The full name of UVA is now provided

What do the “trans arachidonic acids” or other-UVA-induced molecules effect on the recipients? 

            In order to provide information regarding this important question, we have added a new paragraph in the Discussion including new references

Figure 1. The two groups should be explained in legend. The differences of two bars are not clear. Open bar and closed bar may be helpful for the readers.

 We apologize for this lack of information, which was corrected accordingly

Table 1b. What are unit of data?

The units are now provided in the Table legend

 Statistical analysis between 2 groups may be helpful.

            We performed statistical analysis between the two groups and report the results as requested

Figure 2 What is “nAUC” and its unit?

We have now included Abbreviations providing the meaning of this dimensionless and commonly used measure for molecular abundance

Figure 2B recommended to be remade.

 We followed this helpful suggestion and re-designed actually all Figures including former Figure 2B

Round 2

Reviewer 2 Report

Dear authors,

The manuscript really improved following this review round.
Still something to check:
Values of mean clot firmness in figure 1C still do not fit with values in table 1. Please check again.

Best regards

Author Response

We apologize for this error, it was the Table capture which was reversed. Thank you indeed for your careful and helpful review!

Reviewer 3 Report

Authors have sufficiently responded to the comments from reviewers. I have no further comments to revised manuscript.

Author Response

We sincerely thank you for the positive review